# Distinct Effector Programs of Brain-Homing CD8^+^ T Cells in Multiple Sclerosis

**DOI:** 10.3390/cells11101634

**Published:** 2022-05-13

**Authors:** Steven C. Koetzier, Jamie van Langelaar, Marie-José Melief, Annet F. Wierenga-Wolf, Cato E. A. Corsten, Katelijn M. Blok, Cindy Hoeks, Bieke Broux, Beatrijs Wokke, Marvin M. van Luijn, Joost Smolders

**Affiliations:** 1Department of Immunology, Erasmus MC, University Medical Center Rotterdam, 3000 Rotterdam, The Netherlands; s.koetzier@erasmusmc.nl (S.C.K.); j.vanlangelaar@erasmusmc.nl (J.v.L.); j.melief@erasmusmc.nl (M.-J.M.); a.wierenga-wolf@erasmusmc.nl (A.F.W.-W.); 2MS Center ErasMS, Erasmus MC, University Medical Center Rotterdam, 3000 Rotterdam, The Netherlands; c.corsten@erasmusmc.nl (C.E.A.C.); k.blok@erasmusmc.nl (K.M.B.); b.wokke@erasmusmc.nl (B.W.); 3Department of Neurology, Erasmus MC, University Medical Center Rotterdam, 3000 Rotterdam, The Netherlands; 4Neuro-Immune Connections and Repair Lab, Department of Immunology and Infection, Biomedical Research Institute, Hasselt University, 3500 Hasselt, Belgium; cindy.hoeks@uhasselt.be (C.H.); bieke.broux@uhasselt.be (B.B.); 5University MS Center, Hasselt University, 3500 Hasselt, Belgium; 6Neuroimmunology Research Group, Netherlands Institute for Neuroscience, 1105 Amsterdam, The Netherlands

**Keywords:** transcription factors, cytotoxicity, pre-existing and brain-residency

## Abstract

The effector programs of CD8^+^ memory T cells are influenced by the transcription factors RUNX3, EOMES and T-bet. How these factors define brain-homing CD8^+^ memory T cells in multiple sclerosis (MS) remains unknown. To address this, we analyzed blood, CSF and brain tissues from MS patients for the impact of differential RUNX3, EOMES and T-bet expression on CD8^+^ T cell effector phenotypes. The frequencies of RUNX3- and EOMES-, but not T-bet-expressing CD8^+^ memory T cells were reduced in the blood of treatment-naïve MS patients as compared to healthy controls. Such reductions were not seen in MS patients treated with natalizumab (anti-VLA-4 Ab). We found an additional loss of T-bet in RUNX3-expressing cells, which was associated with the presence of MS risk SNP rs6672420 (*RUNX3*). RUNX3^+^EOMES^+^T-bet^−^ CD8^+^ memory T cells were enriched for the brain residency-associated markers CCR5, granzyme K, CD20 and CD69 and selectively dominated the MS CSF. In MS brain tissues, T-bet coexpression was recovered in CD20^dim^ and CD69^+^ CD8^+^ T cells, and was accompanied by increased coproduction of granzyme K and B. These results indicate that coexpression of RUNX3 and EOMES, but not T-bet, defines CD8^+^ memory T cells with a pre-existing brain residency-associated phenotype such that they are prone to enter the CNS in MS.

## 1. Introduction

CD8^+^ memory T cells are highly abundant in postmortem human brain white matter and lesions of patients with multiple sclerosis (MS) [1,2]. These cells generally exert their cytotoxic functions through perforin (PRF) and granzyme B (GZMB) excretion [3]. PRF is a pore-forming protein that targets the cell membrane and paves the way for the serine protease GZMB to induce apoptosis [3]. However, in contrast to peripheral blood, CD8^+^ memory T cells localized within human brain white matter produce a limited amount of GZMB, but high levels of GZMK [4,5]. Next to their atypical cytotoxic profile, CD8^+^ memory T cells residing in the MS brain display a unique CD20^dim^ and CD69^+^ tissue-resident memory (T_RM_) phenotype [4,5,6]. Based on evidence originating from studies with mice, T_RM_ maturation begins within the periphery and is determined by underlying genetic and environmental factors [7]. To date, it remains unclear whether and how such brain-resident memory T cell precursors can be identified in MS patients. Resolving this issue could aid the development of novel therapeutic strategies.

The differentiation and effector program of CD8^+^ memory T cells depends on mutual expression of the transcription factors Runt-related transcription factor 3 (RUNX3), Eomesodermin (EOMES) and the T-box transcription factor (T-bet) [8,9]. Since RUNX3 is a master regulator of both cytotoxicity [9] and T_RM_ formation [10,11], it is tempting to speculate that RUNX3 together with EOMES and/or T-bet also determines the brain-homing potential of CD8^+^ memory T cells in MS patients. Interestingly, the International Multiple Sclerosis Genetics Consortium recently identified a MS risk single-nucleotide polymorphism (SNP) located in the coding region of *RUNX3* (rs6672420, exon 1: Asn18Ile) the function of which has not been described thus far [12,13].

To better understand the programs used by CD8^+^ memory T cells to infiltrate the MS brain, we assessed how RUNX3, EOMES and T-bet (co)expression is related to cytotoxic, tissue-homing and brain residency-associated features using blood, CSF and brain tissues from different MS cohorts, including treatment-naive patients genotyped for MS risk SNP rs6672420 (*RUNX3*). 

## 2. Materials and Methods

### 2.1. Patients, Genotyping and Sampling

Early MS patients were diagnosed based on the McDonald 2017 criteria [14] and included at the MS center ErasMS, Erasmus MC. From these patients, blood and CSF samples were collected for immunophenotyping and were applicable for genetic profiling. Ex vivo blood samples from treatment-naive MS patients were genotyped for rs6672420 according to a method previously published [12]. Furthermore, blood samples were collected from MS patients with a relapsing disease course who clinically responded to natalizumab for 18 months and thus did not experience any neurological attacks. For patients and controls, peripheral blood was collected using CPT tubes (BD Biosciences, Erembodegem, Belgium) containing sodium heparin for cell-based analysis. Peripheral blood mononuclear cells (PBMCs) were isolated according to the manufacturer’s instructions and were used from either fresh or frozen material for immunophenotyping. PBMCs were frozen down in RPMI 1640 with L-Glutamine (Lonza, Verviers, Belgium) containing 20% fetal calf serum (Thermo Fisher Scientific, Landsmeer, The Netherlands) and 10% dimethyl sulfoxide (Sigma-Aldrich, St Louis, MO, USA), and were stored in liquid nitrogen until further use. Postmortem brain white matter tissues were obtained from autopsied donors (Netherlands Brain Bank, Amsterdam, The Netherland) and freshly processed as previously described [1]. Cohort characteristics are summarized in Table 1. No comparisons between fresh and frozen material were performed.

### 2.2. Flow Cytometry

We present the fluorescent-labeled anti-human monoclonal antibodies used in Appendix A. In all experiments, viable cells were analyzed using Fixable Viability Stain 700 (BD Biosciences) or Fixable Viability Dye eFluor 506 (Thermo Fisher Scientific), which were added for 15 min at 4 °C in the dark. Surface markers were stained similarly for 30 min. The Transcription Factor Buffer Set (BD Biosciences) was used to analyze intracellular expression according to the manufacturer’s instructions. In short, cells were fixed for 40 min, after which permeabilization was performed for 10 min and intracellular markers were stained for 40 min, all at 4 °C in the dark. For cytokine measurements, PBMCs were plated at a concentration of 0.5 × 10^6^/mL in RPMI 1640 containing 5% inactivated human AB serum (Sanquin) and Pen/Strep. For cytokine stainings, cells were rested overnight (37 °C) and stimulated with phorbol 12-myristate 13-acetate (50 ng/mL) and ionomycin (1 µg/mL; both obtained from Sigma-Aldrich) for 5 h. GolgiStop (1:1500; BD Biosciences) was added during the last 2.5 h of stimulation and cells were stained according to the protocol mentioned above. Samples were measured using the LSRFortessa and/or sorted using the FACSAria III (both BD Biosciences). CD8^+^ memory (CD45RA^−^) T cells were analyzed using FACSDiva software (version 8.0.2, BD Biosciences).

### 2.3. Transmigration Assay

Positive selection of CD3^+^ T cells from PBMCs was performed prior to sorting using the MojoSort^TM^ Human CD3 selection kit according to the manufacturer’s protocol (BioLegend, San Diego, CA, USA). CD8^+^ memory (CD4^−^CD8^+^CD45RO^+^) T cells were sorted using a FACSAria^TM^ Fusion (BD Biosciences). The human brain endothelial cell line hCMEC/D3 (Tébu-Bio, Le Parray-en-Yvelines, France) was used for Boyden chamber migration assays and was cultured in rat tail type I collagen-coated (Sigma-Aldrich) culture flasks containing EGM-2 MV medium (Lonza) supplemented with 2.5% fetal bovine serum (FBS) (Thermo Fisher Scientific). For migration assays, hCMEC/D3 cells were cultured in 24-well translucent 3 µm Thincerts (Greiner Bio-One, Vilvoorde, Belgium) at a density of 0.25 × 10^5^ cells/cm^2^. On days three and five, hCMEC/D3 cells were replenished with EBM2 (Lonza) supplemented with 10 µg/mL gentamicin, 1 µg/mL amphotericin B, 1 ng/mL fibroblast growth factor, 1.4 µM hydrocortisone (all obtained from Sigma-Aldrich) and 2.5% FBS (Thermo Fisher Scientific). On day six, hCMEC/D3 cells were replenished with EBM2 supplemented with 10 µg/mL gentamicin, 1 µg/mL amphotericin B, 1 ng/mL fibroblast growth factor and 0.25% FBS and were rested for 24 h prior to performing the migration assay. CD8^+^ memory T cells were plated in triplicate at a concentration of 2.4–5.0 × 10^5^/insert. The migration assays were performed for 24 h, after which T cells from the wells (migrated) and from the inserts (non-migrated) were collected, counted and used for flow cytometric analyses. 

### 2.4. Statistics

Statistical analyses were performed using GraphPad Prism (version 9, San Diego, CA, USA); specific details are given in each figure legend. Statistical analyses were corrected for multiple testing using the false discovery rate (FDR) method of Benjamini, Krieger and Yekutieli (BKY). *p*-values of < 0.05 (*) were considered significant.

## 3. Results

### 3.1. RUNX3 and Its Variant rs6672420 Disassociate with T-bet, but Not EOMES Expression in Blood CD8^+^ Memory T Cells

To evaluate the association of RUNX3, EOMES and T-bet with brain-homing T cells in MS, we first compared the presence of these factors in CD8^+^ memory T subsets in the blood of healthy controls (HCs; *n* = 8), treatment-naive MS patients (*n* = 18) and MS patients who clinically responded to natalizumab (NTZ-MS; *n* = 8). By comparing the HC and treatment-naive MS groups, we were able to investigate which T cell subsets tend to recruit to the CNS of MS patients and thus disappear from the circulation in untreated conditions. As natalizumab is a monoclonal antibody against VLA-4 that traps brain-homing T cells within the circulation [15], we expected these cells to be enriched in the blood of NTZ-treated MS patients. We validated this approach earlier for brain-homing CD4^+^ T cell and B cell subsets in MS [16,17]. The proportions of CD45RA^−^ CD8^+^ memory cells within the viable T cell gate were comparable between these cohorts (*p* = 0.948; median: 10.1%, interquartile range (IQR): 8.1–11.5, median: 10.5%, IQR: 8.3–12.5; and median: 10.0%, IQR: 8.0–12.6, for HC, MS and NTZ-MS groups, respectively).

RUNX3 and EOMES, but not T-bet expression was decreased in blood CD8^+^ memory T cells in the treatment-naive MS group versus the HC group (*p* < 0.01 and *p* < 0.001); this was not seen in the NTZ-MS cohort (*p* < 0.05, Figure 1A). Similar observations were made for RUNX3, but not EOMES or T-bet expression levels (Appendix A). A sensitivity analysis excluding the two treatment-naive MS patients receiving a prior steroid pulse yielded no different results (data not shown). At CD8^+^ T cell subset level, both the treatment-naive MS and NTZ-MS groups showed increased frequencies of RUNX3^−^EOMES^−^T-bet^−^ cells when compared to the HC group (*p* < 0.0001 and *p* < 0.05). This was less evident in the NTZ-MS group, which contained a higher proportion of RUNX3^+^EOMES^−^T-bet^−^ cells (*p* < 0.01 and *p* < 0.0001, Figure 1B). In RUNX3-expressing cells, an additional loss in T-bet was found for both the treatment-naive MS and NTZ-MS groups (*p* < 0.05), leaving RUNX3^+^EOMES^+^T-bet^−^ cell frequencies unaffected (Figure 1B). Similar results were obtained for treatment-naive MS patients when comparing carriers and non-carriers of MS risk SNP rs6672420 (*RUNX3*). The presence of rs6672420 was associated with increased proportions of RUNX3^−^EOMES^−^T-bet^−^ (*p* < 0.01) and RUNX3^+^EOMES^−^T-bet^−^ (*p* < 0.05) cells, while RUNX3 and T-bet-coexpressing subsets were markedly reduced (*p* < 0.05) within the CD8^+^ memory T cell pool (Figure 1C,D). The proportions of RUNX3^+^EOMES^+^T-bet^−^ cells were not different between risk and non-risk carriers (Figure 1D). Notably, no differences were observed when exploring these associations with the presence of rs438613 or rs13327021, two intergenic MS risk SNPs located close to *EOMES* (Appendix A) [12].

Overall, these data demonstrate that the coexpression of RUNX3 with EOMES and T-bet is aberrant in MS, which seems to be at least partly influenced by the presence of rs6672420 (*RUNX3*). Given the observed impact of NTZ treatment, such aberrances may be associated with the preferential recruitment of distinct CD8^+^ memory T cell subsets to the CNS of MS patients.

### 3.2. Circulating RUNX3^+^EOMES^+^T-bet^−^ Memory CD8^+^ T Cells Show a Discriminative, Brain-Homing Phenotype 

To study how transcription factor expression patterns contribute to the brain-homing potential of CD8^+^ memory T cells, we first analyzed the surface expression of the brain-homing markers CCR5, CCR6 and CXCR3 in this context. In the treatment-naive MS group, CCR5 was strongly upregulated on RUNX3^+^ cells expressing EOMES and particularly in those without T-bet (Figure 2A). A lack of T-bet also corresponded to higher CCR6 (Figure 2B) and CXCR3 (Figure 2C) surface levels in CD8^+^ memory T cells. Since brain-homing potential is probably also defined by cytotoxic capacity, we next compared the intracellular expression of cytotoxic and pro-inflammatory molecules between these subsets. Similar to CCR5, GZMK (Figure 2D) was most abundant in RUNX3^+^EOMES^+^T-bet^−^ cells. Conversely, both GZMB and PRF (Figure 2E,F) were upregulated in the presence of T-bet rather than EOMES (Figure 2E,F). The same differences in effector phenotypes were found when analyzing the HC and NTZ-MS groups (Appendix A), indicating that these effects are not restricted to untreated MS patients. Furthermore, RUNX3^+^ cells that expressed EOMES upregulated IFN-γ and TNF-α, but not IL-17A expression, which, in contrast to brain-homing markers and GZMK (Figure 2A–D), was not different when compared to RUNX3^−^ cells (Appendix A). 

Hence, CD8^+^ memory T cells expressing RUNX3 together with EOMES and not T-bet display a broad repertoire of MS-associated brain-homing and cytotoxic markers, which further supports the hypothesis that this subset is preferably recruited to the MS brain.

### 3.3. RUNX3^+^EOMES^+^T-bet^−^Memory CD8^+^ T Cells Are Enriched and Display a CD20^dim^ CD69^+^ Brain Residency-Associated Phenotype in Early MS CSF

To confirm the observed indications for preferential brain-homing, we first looked into these subsets in the paired blood and CSF samples of seven MS patients (Table 1). In accordance with earlier work [18], we found that CD45RA^−^ CD8^+^ memory T cells made up a minority of the viable T cell fraction in the CSF (median: 16.2%, IQR: 14.8–20.7). Indeed, we found that the RUNX3^+^EOMES^+^T-bet^−^ subset was selectively enriched and that it dominated the CD8^+^ memory T cell pool in the CSF of early MS patients (*p* < 0.01, Figure 3A,B). Previously, others showed that CSF T cells were enriched for CD20^dim^ expression and CD69^+^ T cell fractions, which have since been associated with CD8^+^ T_RM_ pools residing in MS brain white matter of normal appearance and in lesions [4,5,6]. Similarly, we observed an enrichment of these markers on CD8^+^ memory T cells in MS CSF compared to paired blood samples (*p* < 0.05, Figure 3C,D). Both CD20^dim^ and CD69^+^ versus CD20^−^ and CD69^−^ CD8^+^ memory T cells, which were localized in the MS CSF, were enriched for the RUNX3^+^EOMES^+^T-bet^−^ profile (*p* < 0.0001 and *p* < 0.05; Figure 3E,F). Notably, this phenotype was already identifiable in CD20^dim^CD8^+^ memory T cells localized in the blood of these patients (*p* < 0.0001; Figure 3E), while CD69^+^ counterparts primarily expressed T-bet (*p* < 0.05 and *p* < 0.01; Figure 3F). 

Consistently, CD20^dim^CD8^+^ memory T cells primarily produced GZMK, which showed a slight increase in MS CSF versus blood (Figure 3G). In accordance with the observed effects of RUNX3 and EOMES coexpression (Figure 2D and Figure 3F), CD69^+^ CD8^+^ memory T cells localized in the CSF expressed more GZMK (*p* < 0.01), but less PRF and GZMB as compared to blood (*p* < 0.01 and *p* < 0.05) (Figure 3G). To confirm the selective transmigratory capacity of these cells towards the CSF, we performed an in vitro transmigration assay using human brain endothelial cells to mimic the blood–brain barrier. Using this assay, we showed a high enrichment of both GZMK and CD69 CD8^+^ memory T cells in the fraction migrated through the brain endothelial cell layer compared to the non-migrated fraction (see Appendix A). Within the CSF, both CD20^dim^ and CD69^+^ CD8^+^ memory T cells expressed more GZMK in comparison to CD20^−^ and CD69^−^ counterparts (*p* < 0.001; Figure 3H). 

We isolated mononuclear cells from postmortem MS brain samples. In accordance with earlier studies [19], we observed a numerically larger proportion of viable T cells to display a CD8^+^CD45RA^−^ memory phenotype (median: 50.9%, IQR: 25.0–72.9) compared to CSF samples of living donors (see above). In line with our previous work showing that T-bet and not EOMES was expressed by CD8^+^ T_RM_ cells localized within the brain [4], we found that RUNX3^+^EOMES^−^T-bet^+^ cells dominated the CD8^+^ memory T cell pool in eight MS brain tissues (Table 1 and Appendix A). While almost all CD8^+^ memory T cells in the MS brain expressed CD69, CD20 was variably expressed (*p* < 0.05, Figure 3I,J). Notably, RUNX3, EOMES and T-bet expression profiles did not differ between CD20^−^ and CD20^dim^ fractions (Figure 3K). In contrast to early MS CSF, CD8^+^ memory T cells isolated from postmortem MS brain tissue mainly coproduced GZMK and GZMB (Appendix A), which was increased in CD20^dim^ counterparts (*p* < 0.01 and *p* < 0.0001, Figure 3L).

In conclusion, the CD8^+^ memory T cell pool in MS CSF was found to be enriched for RUNX3^+^EOMES^+^T-bet^−^ cells which display a GZMK-expressing CD20^dim^CD69^+^ brain residency-associated phenotype and thus potentially represent brain CD8^+^ T_RM_ precursors. 

## 4. Discussion

Human CD8^+^ T_RM_ cells are involved in different types of pathologies and are thought to have precursors [7,20,21], but so far this population has not been identified within the circulation of patients with chronic neuroinflammatory disorders such as MS. Identifying this CD8^+^ T_RM_ precursor and its driving factors will be an important step towards the discovery of new biomarkers and the development of novel therapeutics for MS. Currently, there is no treatment option available that can completely stop disease progression in these patients [22]. In this study, we provide the first clues as to the existence of such a population of CD8^+^ T cells with a distinct effector program, which might be driven by underlying genetic factors, including the MS risk SNP rs6672420 in *RUNX3* (rs6672420).

We have shown that MS patients have lower frequencies of RUNX3- and EOMES-expressing CD8^+^ memory T cells in peripheral blood, which is rescued by natalizumab treatment. In addition, circulating T-bet-coexpressing populations were generally decreased in all MS cohorts, leaving RUNX3^+^EOMES^+^T-bet^−^ frequencies generally intact. This phenotype seemed to relate to the presence of the rs6672420 risk allele in treatment-naive MS patients. There has been one study that reported that this SNP affects the methylation of RUNX3 and thereby reduces its expression [12,13]. While we did not observe this, possibly due to differences in cell-intrinsic methylation patterns, more in-depth investigations are needed to elucidate how rs6672420 impacts RUNX3 in CD8+ memory T cells. Since the presence of rs6672420 was associated with altered T-bet coexpression according to our data, these studies should primarily focus on protein–protein interactions. It has been found that RUNX3 regulates the expression of both EOMES and T-bet [9,23]. However, it was shown that high T-bet expression is prevented by RUNX3 [9] and is a prerequisite for maturation into a CD8^+^ T_RM_ cell [10]. This is consistent with our results that especially circulating RUNX3^+^EOMES^+^T-bet^−^ cells but not RUNX3^+^EOMES^−^T-bet^+^ cells display a brain-homing and an MS brain residency-associated memory CD8^+^ T cell phenotype, as reflected by increased expression of CCR5 [4,24,25], CCR6 [26], CXCR3 [25], GZMK [2,4], CD20 [6] and CD69 [4,5], as well as their enrichment in MS CSF. It should be noted that, EOMES was found to downregulate HOBIT in mice [27], a transcription factor generally considered a hallmark of T_RM_ cells [28,29]. However, and similar to other CD8^+^ T_RM_-associated markers such as CD103, the expression of this molecule in human T_RM_ cells is tissue-dependent [7] and HOBIT is absent in those populations that reside within the MS brain [4]. Therefore, it is still likely that RUNX3^+^EOMES^+^T-bet^−^ cells are precursors of brain CD8^+^ T_RM_-populations in MS; however, their differentiation is likely to be controlled by other T_RM_-associated transcription factors, such as B lymphocyte-induced maturation protein-1 (BLIMP-1) [28]. The fact that total RUNX3^+^ and EOMES^+^ CD8^+^ memory T cells, but not RUNX3^+^EOMES^+^T-bet^−^ coexpressing cells, accumulated in the circulation of MS patients after natalizumab treatment indicates that other transcription factors such as BLIMP-1 could be especially relevant to better define this brain-homing subset. Together with the fact that BLIMP1 expression is driven by RUNX3 [9], this should be further investigated in subsequent studies.

In addition, we cannot exclude the possibility that circulating RUNX3-expressing CD8^+^ memory T cells lacking both T-bet and EOMES also contribute, as precursors, to the perivascular CD8^+^ T_RM_ pool in MS. Our results showed that the RUNX3^+^EOMES^−^T-bet^−^ subset within the CD8^+^ memory T cell pool is increased in the blood of MS patients carrying rs6672420 and in those patients treated with natalizumab. The latter result points to their migration into the CNS under treatment-naive conditions, though we did not find that these cells were enriched in the CSF of MS patients. Although lower in expression as compared to the RUNX3^+^EOMES^+^T-bet^−^ fraction, these RUNX3^+^EOMES^−^T-bet^−^ cells do express the brain-homing markers CCR6 [26,30] and CXCR3 [25]. Therefore, it is possible that EOMES coexpression and thereby the expression of associated markers, such as CCR5 and GZMK, is induced after interaction with the blood–brain barrier or shortly after entry into the CSF. However, it is feasible that certain CD8^+^ memory T cell subsets had already left the circulation and entered the CNS or are functionally affected by natalizumab in MS patients, therefore influencing our findings.

Lastly, we confirmed that within the MS brain, EOMES is downregulated and T-bet expression is upregulated in brain CD8^+^ T cells, which is consistent with an earlier study [4]. Here, it was shown that T-bet expression is lower in brain CD8^+^ T_RM_ cells when compared to blood counterparts [4]. This is consistent with another study, which found that although high T-bet expression hampers T_RM_ formation through downregulation of the CD103-driving transforming growth factor beta receptor [31], low T-bet expression is required for the expression of the IL-15 receptor [7,32]. This not only promotes the survival of these T_RM_ cells but is also required for their formation, as it induces BLIMP-1 expression together with RUNX3 [9,33]. As we and others have investigated late-stage postmortem MS brain tissues from elderly individuals, the impact of age-associated T cell senescence or effects of an advanced MS state on our findings must not be overlooked.

In this study, we have demonstrated that circulating RUNX3^+^EOMES^+^T-bet^−^ CD8^+^ memory T cells display a phenotype that closely resembles CD8^+^ T_RM_ cells residing within the MS brain. These results should be validated using larger, less heterogeneous and preferably completely untreated patient cohorts and exploited for more in-depth analysis of transcriptional and phenotypic similarities at the single-cell level. This should include an analysis to assess whether the usage of disease-modifying treatments, as used by MS patients in this study, affect the obtained results. Detailed characterization of pre-existing circulating CD8^+^ T cells with a brain residency-associated phenotype could pave the way for the development of new therapeutic approaches to modulate the contribution of brain CD8^+^ T_RM_ cells to early but also advanced progressive MS.

## Figures and Tables

**Figure 1 cells-11-01634-f001:**
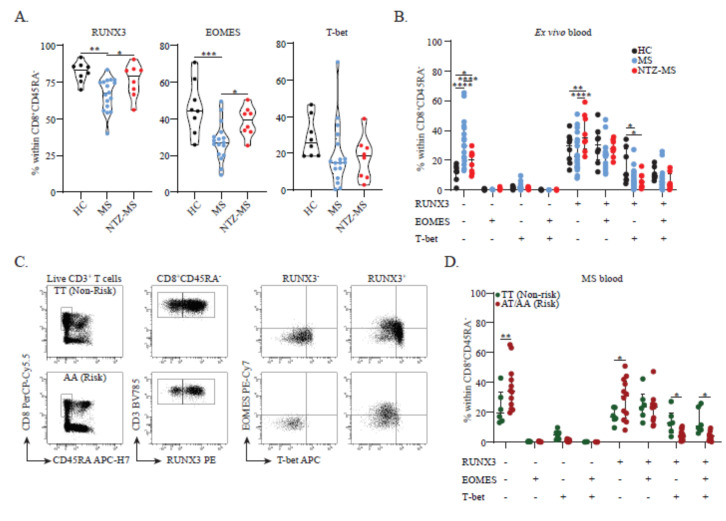
RUNX3, EOMES and T-bet expression patterns and the impact of rs6672420. (**A**) Expression of RUNX3, EOMES or T-bet expression in the CD8^+^ memory T cell pool (HC, *n* = 8; MS, *n* = 18; and NTZ-MS, *n* = 8). (**B**) Coexpression of RUNX3, EOMES and T-bet in the CD8^+^ memory T cell pool (HC, *n* = 8; MS, *n* = 18; and NTZ-MS, *n* = 8). (**C**) Representative FACS plot showing the distribution of CD8^+^ memory T cells (left), their RUNX3 expression (middle) and EOMES and T-bet coexpression (right) within a blood sample of a MS patient who was either homozygous non-risk (top, TT) or a risk carrier (bottom, AA) for rs6672420. (**D**) Coexpression of RUNX3, EOMES and T-bet in the peripheral blood CD8^+^ memory T cell pool of MS patients that did (*n* = 12, *n* = 6 AT and AA) or did not (*n* = 6, TT) carry the rs6672420 risk allele. Data were compared using a two-way ANOVA or (**B**,**D**) Kruskal–Wallis tests, both with FDR-BKY corrections. * *p* < 0.05, ** *p* < 0.01, *** *p* < 0.001 and **** *p* < 0.0001. “HC” = healthy controls, “MS” = treatment-naive MS patients, “NTZ-MS” = MS patients who clinically responded to natalizumab treatment for 18 months.

**Figure 2 cells-11-01634-f002:**
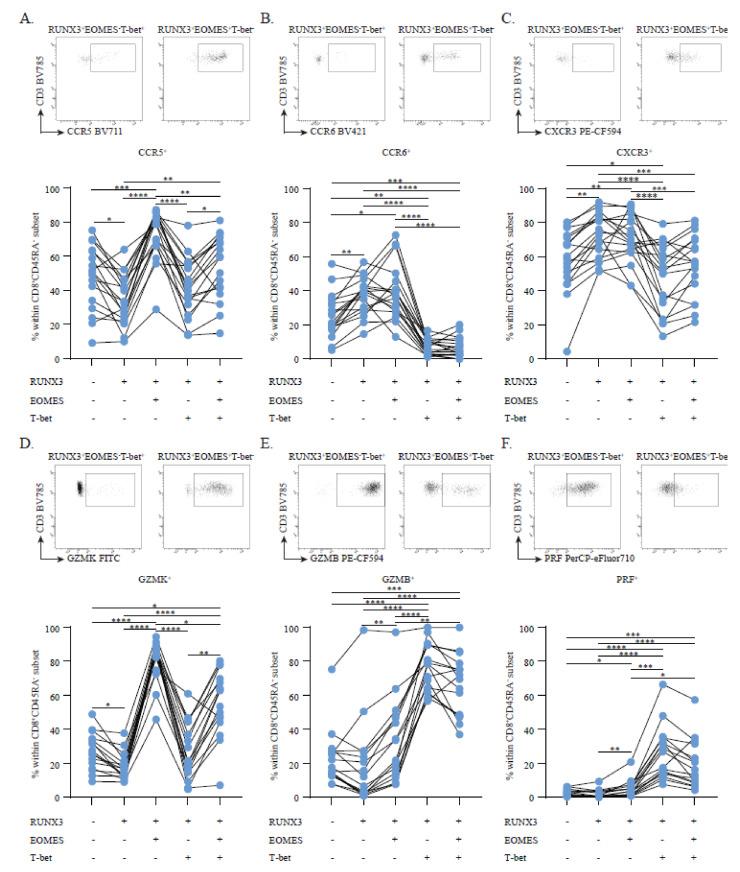
The influence of RUNX3, EOMES and T-bet on brain-homing and cytotoxic marker expression. CCR5 (**A**), CCR6 (**B**), CXCR3 (**C**), GZMK (**D**), GZMB (**E**) and PRF (**F**) expression in the CD8^+^ memory T cell pool of treatment-naive MS patients (*n* = 16–18) with RUNX3, EOMES and T-bet (co)expression. Above each graph a representative FACS plots shows the difference in marker expression between RUNX3^+^ cells expressing T-bet (left) and EOMES (right). Lines represent paired measurements from the same donor. Data were compared using a two-way ANOVA with FDR-BKY correction tests. * *p* < 0.05, ** *p* < 0.01, *** *p* < 0.001 and **** *p* < 0.0001.

**Figure 3 cells-11-01634-f003:**
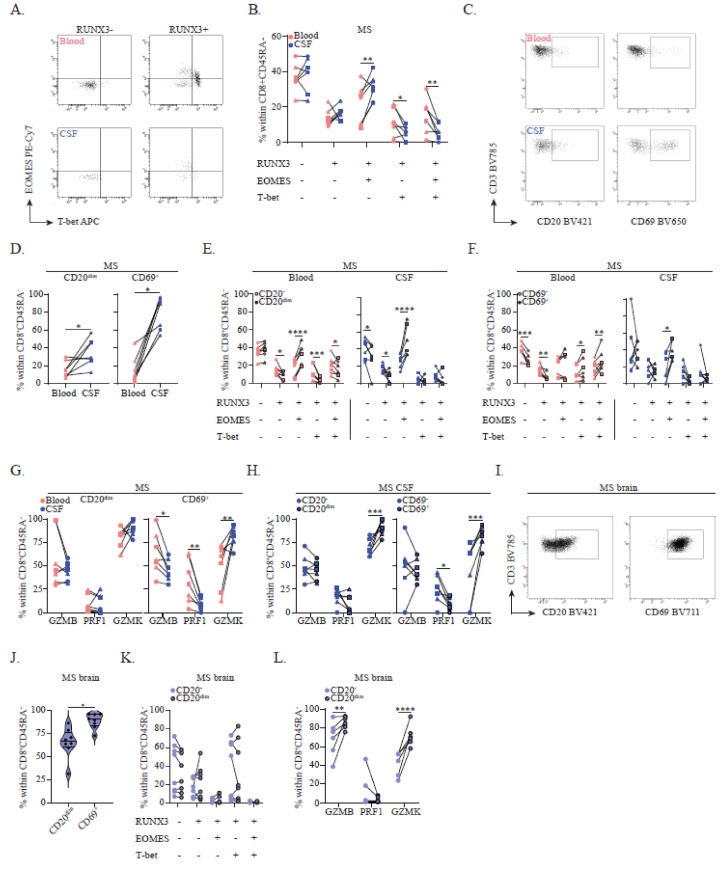
The relation between RUNX3, EOMES and T-bet expression patterns and brain residency-associated marker expression in MS CSF and brain tissue. (**A**) Representative FACS plot showing the distribution of EOMES and T-bet within the RUNX3^−^ and RUNX3^+^ CD8^+^ memory T cell pool in paired early MS blood and CSF and its quantification (**B**) (*n* = 7). (**C**) Representative FACS plot showing CD20 and CD69 expression on CD8^+^ memory T cells in paired early MS blood and CSF. (**D**) Frequencies of CD20^dim^ and CD69^+^ cells within the CD8^+^ memory T cell pool in paired early MS blood and CSF (*n* = 7). (**E**) Percentages of paired CD20^−^ versus CD20^dim^ and (**F**) CD69^−^ versus CD69^+^ cells with RUNX3, EOMES and T-bet (co)expression in the CD8^+^ memory T cell pool in early MS blood and CSF (*n* = 7). (**G**) GZMB, PRF and GZMK expression by CD20^dim^ and CD69^+^ CD8^+^ memory T cells in paired early MS blood and CSF (*n* = 7). (**H**) Percentages of GZMB, PRF and GZMK expression by paired CD20^−^ versus CD20^dim^ and CD69^−^ versus CD69^+^ CD8^+^ memory T cells in early MS CSF (*n* = 7). (**I**) Representative FACS plot showing CD20 and CD69 expression on CD8^+^ memory T cells in late-stage MS brain tissue and its quantification (**J**) (*n* = 8 of 4 brain donors). (**K**) Frequencies of paired CD20^−^ versus CD20^dim^ cells with RUNX3, EOMES and T-bet (co)expression in the CD8^+^ memory T cell pool in late-stage postmortem MS brain tissue (*n* = 8 of 4 brain donors). (**L**) GZMB, PRF and GZMK expression by paired CD20^−^ versus CD20^dim^ CD8^+^ memory T cells localized in late-stage postmortem MS brain tissue (*n* = 8 of 4 brain donors). Lines represent paired measurements for the same donor. (**B**,**D**–**F**) Treatment-naive patients were depicted as circles and patients treated with methylprednisolone or disease-modifying therapy were depicted as squares or triangles, respectively. Data were compared using the Wilcoxon rank-sum test or (**D**,**J**) two-way ANOVAs with FDR-BKY correction. * *p* < 0.05, ** *p* < 0.01, *** *p* < 0.001 and **** *p* < 0.0001.

**Table 1 cells-11-01634-t001:** Overview of cohort characteristics.

Peripheral Blood	Healthy	MS (No Tx) ^b^	NTZ-MS ^a,b^
Individuals, n	22	18	18
Females, n (%)	12 (55)	9 (50)	9 (50)
Age in years, median (range)	38 (22–63)	39 (24–48)	35 (19–62)
Disease duration in months, median (range) ^c^	NA	2 (0–7)	89 (32–256)
MP < 1 month prior to sampling, n (%)	NA	2 (11)	NA
**CNS**	Paired early multiple sclerosis blood and CSF ^b^	Late-stage postmortem multiple sclerosis brain tissue ^f^	
Individuals, n	7	4	
Females, n (%)	3 (43)	3 (75)	
Age in years, median (range)	41 (20–62)	73 (66–77)	
Multiple sclerosis, n (%)	7 (100)	4 (100)	
Disease duration in months, median (range) ^c^	11 (9–315)	NA	
MP < 1 month prior to sampling, n (%)	2 (29)	NA	
Other treatment prior to sampling, n (%) ^d^	3 (43)	NA	
Postmortem delay in hours, median (range) ^e^	NA	07:00 (06:50–07:04)	

Abbreviations: Healthy = healthy individuals and controls; MS (No Tx) = treatment-naive multiple sclerosis; NTZ-MS = multiple sclerosis patients who were treated for 18 months with and clinically responded to natalizumab treatment; MP = methylprednisolone; NA = not applicable. ^a^ Cohorts were used for transcription factor, granzyme and chemokine receptor expression screens, cytokine-production screens or transmigration assays (see exact numbers in figure legends). ^b^ Diagnosis according to the McDonald 2017 criteria. ^c^ Time from clinically isolated syndrome diagnosis to sampling. ^d^ Dimethyl fumarate, Interferon β1a or glatiramer acetate. ^e^ Given as hours:minutes. ^f^ Disease subtype: SPMS (*n* = 2), PPMS (*n* = 2); treatment: none (*n* = 1), IFN-β (*n* = 2), azathioprine (*n* = 1); cause of death: legally granted euthanasia (*n* = 2), dehydration (*n* = 2).

## Data Availability

The data presented in this study are available upon reasonable request.

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
