# Peer review of "Distinct Effector Programs of Brain-Homing CD8+ T Cells in Multiple Sclerosis"

_cells, 2022, doi:10.3390/cells11101634_

Round 1
Reviewer 1 Report
In the manuscript entitled “Distinct Effector Programs of Brain-Homing CD8+ T Cells in Multiple Sclerosis”, Koetziger et al assessed how expression of the transcription factors RUNX3, EOMES and T-bet in memory CD8+ T cells related to cytotoxic, tissue-homing and brain residency-associated features in blood, CSF and brain tissues in MS patients. Data from their analyses indicates that co-expression of RUNX3 and EOMES defines CD8+ memory T cells that are prone to enter CNS in MS patients. Their results might pave the way to develop novel medicine regimes to improve MS by targeting factors important for CD8 TRM development.
This is a well-grounded story and nicely written paper. It build upon previous research performed by the research group. I have only minor concerns:
There are many molecules that are measured and hard to keep track on everything. The manuscript would benefit from an illustration describing the approaches and molecules involved in CD8 TRM development.
This rs 6672420 SNP is intronic, but according to GTEx not identified as an eQTL. The potential mechanisms beyond the importance of this SNP for CD8 TRM development could be elaborated on in the discussion.
Throughout the result section, it is stated that expression of the transcription factors are decreased or increased. However, the figures show the percentage of cells expressing the different transcription factors. This is a bit confusing. Furthermore, does the expression level of the transcription factors give additional information that could be of value for the phenotyping of the studied cell type? For instance, in cases where there are no difference in the percentage of a given cell subtype, are there differences in the expression levels of RUNX3, EOMES or T-bet?
In figure 1A, T-bet expression. Is the lack of significant difference between HC and MS due to one outlier in the MS group?
In figure 3J, is this significant finding driven by one outlier?
In the figure legends, please specify in which comparisons the different statistical tests were used.
Please rephrase the term “relatively intact” in line 144 in the result section.
In the Materials and Methods section it is stated that PBMCs were used either fresh or frozen for immunophenotyping. It is unclear whether there is a mixture of fresh and frozen samples that are compared in the same experiment. If so, data needs to be provided showing the freezing/thawing procedure does not affect the parameters measured by immunophenotyping.
Please provide the concentration of PMA and Ionomycin used (section 2.2).
Remember passive past in the Materials and methods, also in the statistics section (section 2.4)
In Table 1, number of individuals and females etc should be written on the form “22” and not as “22.0” etc
Author Response
Reviewer 1
In the manuscript entitled “Distinct Effector Programs of Brain-Homing CD8+ T Cells in Multiple Sclerosis”, Koetziger et al assessed how expression of the transcription factors RUNX3, EOMES and T-bet in memory CD8+ T cells related to cytotoxic, tissue-homing and brain residency-associated features in blood, CSF and brain tissues in MS patients. Data from their analyses indicates that co-expression of RUNX3 and EOMES defines CD8+ memory T cells that are prone to enter CNS in MS patients. Their results might pave the way to develop novel medicine regimes to improve MS by targeting factors important for CD8 TRM development.
This is a well-grounded story and nicely written paper. It build upon previous research performed by the research group. I have only minor concerns:
We would like to thank the reviewer for carefully reading this manuscript, his compliments on our work and giving valuable feedback.
There are many molecules that are measured and hard to keep track on everything. The manuscript would benefit from an illustration describing the approaches and molecules involved in CD8 TRM development.
As suggested by the editorial office and this reviewer, we now added this as a graphical abstract to the paper.
This rs 6672420 SNP is intronic, but according to GTEx not identified as an eQTL. The potential mechanisms beyond the importance of this SNP for CD8 TRM development could be elaborated on in the discussion.
Rs6672420 is a coding SNP located in exon 1 (https://www.ncbi.nlm.nih.gov/snp/rs6672420), we now clarified this in the introduction and elaborated on this in the discussion section.
Throughout the result section, it is stated that expression of the transcription factors are decreased or increased. However, the figures show the percentage of cells expressing the different transcription factors. This is a bit confusing. Furthermore, does the expression level of the transcription factors give additional information that could be of value for the phenotyping of the studied cell type? For instance, in cases where there are no difference in the percentage of a given cell subtype, are there differences in the expression levels of RUNX3, EOMES or T-bet?
We agree with the reviewer that expression levels could also be impacted, therefore we assessed RUNX3, EOMES and T-bet surface expression (MFI) on each positive CD8+ memory T subset from healthy controls, MS patients and natalizumab-treated MS patients and added this as a supplementary figure. We found similar results as found for percentages for RUNX3 and T-bet, but not EOMES. This indicates that RUNX3 expression are additionally impacted next to differences in % of RUNX3+ memory CD8+ T cells. As requested by this reviewer, we now added these results as a supplementary figure and added this to the result section.
In figure 1A, T-bet expression. Is the lack of significant difference between HC and MS due to one outlier in the MS group?
We could not find a valid experimental or medical reason to exclude this patient. Notably, the outlier patient was not the same between RUNX3, EOMES and T-bet analysis. Statistical reanalyses revealed that only RUNX3 trended in the same direction but became non-significant between MS and NTZ-MS when the lowest value was excluded (p = 0.08). The difference between HC and MS remained.
In figure 3J, is this significant finding driven by one outlier?
We could not find a valid experimental or medical reason to exclude this patient. Statistical reanalyses censoring this one case remained significant (p = 0.04).
In the figure legends, please specify in which comparisons the different statistical tests were used.
We now adapted this throughout the manuscript.
Please rephrase the term “relatively intact” in line 144 in the result section.
We now adapted this in the manuscript.
In the Materials and Methods section it is stated that PBMCs were used either fresh or frozen for immunophenotyping. It is unclear whether there is a mixture of fresh and frozen samples that are compared in the same experiment. If so, data needs to be provided showing the freezing/thawing procedure does not affect the parameters measured by immunophenotyping.
We did not compare fresh with frozen material. We now clarified this in the methods section.
Please provide the concentration of PMA and Ionomycin used (section 2.2).
We now adapted this in the manuscript.
Remember passive past in the Materials and methods, also in the statistics section (section 2.4)
We now adapted this in the methods section.
In Table 1, number of individuals and females etc should be written on the form “22” and not as “22.0” etc
We now adapted this in the methods section.

Reviewer 2 Report
The authors of the manuscript performed an analysis of memory CD8+ T cells in MS patients with special focus laid on parameters associated with brain homing of these cells. The topic of the study is definitely interesting and important for better understanding of immune-pathology of MS, however there are some major issues which have to be solved before the work can be fully assessed.
- The information about ethical bord approval of all the procedures is missing. The authors should provide detailed information, including the agreement for genetic examination.
- The study groups are small and heterogenous, especially the range of age arises doubts about the possible age-dependent differences of immune parameters. Moreover, the small groups are further divided for particular experiments. Hence the authors have to prove that there are no differences in demographic characteristics of groups used for EACH particular experiments.
- Another problem is the very small number and heterogeneity of “paired blood-CSF sample group”. Not only age but also disease duration differ significantly among the patients in this group. Additionally 3 out of 7 patients were treated with different DMTs which with great probability influenced the immune cells in blood and possibly also in CSF. Moreover 2 out of 7 patients had received methylprednisolone shortly before the blood/CSF analysis, which not only means that the results were influenced by the very strong action of steroids but also implicates that the patients were examined during MS exacerbation, adding another confounder in the analysis. Thus, these patients should be excluded from the analysis. The same issue with methylprednisolone refers to the “treatment naïve MS” group.
- The information about clinical course of MS in autopsied donors is not sufficient. Their clinical course and treatment history is very important for any comparisons. The authors should also prove that the significant difference in age does not hamper the inclusion of these data in the analysis.
Author Response
Reviewer 2
The authors of the manuscript performed an analysis of memory CD8+ T cells in MS patients with special focus laid on parameters associated with brain homing of these cells. The topic of the study is definitely interesting and important for better understanding of immune-pathology of MS, however there are some major issues which have to be solved before the work can be fully assessed.
We would like to thank the reviewer for carefully reading the manuscript
The information about ethical bord approval of all the procedures is missing. The authors should provide detailed information, including the agreement for genetic examination.
We supplied the missing information to the editor and adapted the manuscript accordingly (see Institutional Review Board Statement).
The study groups are small and heterogenous, especially the range of age arises doubts about the possible age-dependent differences of immune parameters. Moreover, the small groups are further divided for particular experiments. Hence the authors have to prove that there are no differences in demographic characteristics of groups used for EACH particular experiments.
We agree with the reviewer that the cohort size is a limitation of this study and added this important point to the discussion section of our manuscript. However, the age difference between the MS CSF and brain tissue cohorts is due to the differences in material and disease course. Brain biopsies are not regularly performed in living MS patients and thus the material are obtained postmortem from progressive MS patients. As we did not directly compare these two cohorts, we think that we presented the data correctly. All relevant demographic characteristics of the cohorts investigated are included in table 1, showing a similar sex and age distribution in the cohorts which are directly compared. This point is discussed extensively in the discussion section (line 315-317).
Another problem is the very small number and heterogeneity of “paired blood-CSF sample group”. Not only age but also disease duration differ significantly among the patients in this group. Additionally 3 out of 7 patients were treated with different DMTs which with great probability influenced the immune cells in blood and possibly also in CSF. Moreover 2 out of 7 patients had received methylprednisolone shortly before the blood/CSF analysis, which not only means that the results were influenced by the very strong action of steroids but also implicates that the patients were examined during MS exacerbation, adding another confounder in the analysis. Thus, these patients should be excluded from the analysis. The same issue with methylprednisolone refers to the “treatment naïve MS” group.
Although we agree with the reviewer that MP and/or DMT usage could impact our results, we did not observe this in our data. However, to accurately present our data we indicated which patients are MP- (square) or DMT-treated (triangle).
The information about clinical course of MS in autopsied donors is not sufficient. Their clinical course and treatment history is very important for any comparisons. The authors should also prove that the significant difference in age does not hamper the inclusion of these data in the analysis.
We reviewed the donor files of the included brain donors and included the information requested by this reviewer in the footnotes of table 1. Please find our response to the comparability of different cohort included above.
Round 2
Reviewer 2 Report
In the revised version of the manuscript the authors did not acknowledge any of the methodological issues indicated in the previous review. It has to be regarded as especially surprising that the authors consider the influence of corticosteroids, DMTs and/or disease exacerbation as unimportant for the solidity of the presented data. The modification of graphical presentation does not prove or exclude the influence of treatment on the parameters. Multiple sclerosis is a not a rare disease and it is a commonly accepted and demanded standard in the research to work on maximally homogenous patient groups including patients of similar age, sex, disease course, treatment status etc. The explanation regarding the demographic characteristics of the study groups is also not sufficient – in the particular experiments the authors used material from subgroups of patients, and the demographics of subgroups was not compared.
Unfortunately, the opinion about the manuscript cannot be changed.
Author Response
As also indicated in our previous reply, we agree with the reviewer that MP and/or DMT usage could impact our findings, so considering this as a limitation of our work is important.
However, because we rely on fresh material coming from irregular lumbar punctures and unique brain autopsies of MS donors, it is commonly accepted in the field that it is at least a challenge to acquire homogeneous and completely treatment-naive patient groups for these kinds of analyses. On this basis, we performed the (in our view) best possible actions to minimize the potential effect of age, sex, disease course and treatment status on results. Additionally, our current findings are in accordance with the presence of CD8+ memory T cells in the compartments analyzed in this study with previous work. This is summarized below.
First, for our blood screens (Fig. 1-2), we pre-selected and matched HC, MS and NTZ-MS samples based on age and sex (Table 1). We acknowledged the difference in disease duration, because of the relevance of analyzing CD8+ T-cell effector programs in both very early disease stages and following treatment with NTZ, which causes peripheral accumulation of brain-homing subsets (main scope of the study). In the NTZ-MS group, all 18 patients had a relapsing disease course (now added to the methods section), and were studied at 18 months and did not experience any relapse during NTZ treatment (L68-70). When we excluded the 2 MP-treated patients from the MS group (Table 1), the differences between groups remained unaffected. We now disclose this in the manuscript (L152-154).
Second, for our CNS compartment screens (Fig. 3), only paired analyses for fresh CSF vs blood samples as well as CD20dim vs CD20- and CD69+ vs CD69- populations in MS CSF and brain tissues were carried out. To deduct the impact of having more naive cells in the circulation, which is also affected by treatment and age (Thome et al., Sci Immunol 2016), we only analyzed CD45RA- memory fractions throughout the study. The now mention this more explicitly throughout the manuscript. The influence of MP and/or DMT should be reduced by comparing effector programs of subsets within the same tissue, which is supported by the same trends found for treated and non-treated samples (Fig. 3).
Third, now included in the text of the respective results sections, we found relatively low percentages of CD8+ memory T cells in MS blood and CSF, while being largely enriched in brain tissues (Table 1), which is consistent with earlier findings by our group and others (e.g. Stüve et al., Arch Neurol 2006; Smolders et al., Semin Immunopathol 2022) and further supports the analysis of their effector programs in the current study.